

# Tencent Meeting forensics based on memory reverse analysis

Shilong Yu[1], Binglong Li[1], Lin Zhu[1], Heyu Zhang[2], Sen Yang[1], Zhangxiao Li[1] and Wenzheng Feng[1]

[1] School of Cryptographic Engineering, Information Engineering University, Zhengzhou, China
[2] School of Cyber Science and Engineering, Huazhong University of Science and Technology, Wuhan, China

## ABSTRACT

Tencent Meeting, an instant meeting software, is widely used at present, but no research has been conducted on its forensics. Since the real-time data generated by such software during meetings will not be stored in the computer disk, the traditional disk forensics method against such software is no longer applicable and needs to obtain evidence through memory analysis. To extract meeting data transmitted during meetings, this article proposes a method for Tencent Meeting forensics based on memory reverse analysis. First, by analyzing the process storage and metadata format of Tencent Meeting in memory, an inverse metadata extraction algorithm is designed. Then, by analyzing the data structure of Tencent Meeting in memory, a meeting data stream engraving algorithm is developed. Finally, the experimental results indicate that the proposed method can effectively extract metadata information such as meeting time, meeting number, topic, and data flow information such as participants, message records, as well as transmitted files from the memory of Tencent Meeting, providing crucial digital evidence for digital crime investigation. Compared with other forensic analysis methods for instant meeting software, our proposed forensic method for Tencent Meeting conducts memory reverse analysis with the entire memory file, enabling the extraction of more comprehensive and abundant forensic data.

# INTRODUCTION

With the rapid development of the digital era, instant meeting software such as Tencent Meeting, DingDing, Google Meet and Zoom have become crucial tools in business communication, remote collaboration, online education, home office, and other fields (*Nicoletti & Bernaschi, 2021*; *Khalid et al., 2022*). Nevertheless, due to the popularity of instant meeting software, it is vulnerable to various network attacks, such as conference bombing, spreading malicious links or files containing viruses in chats, and stealing conference links and permissions. Some researchers believe that the relevant data in the operation of instant meeting software is the key to solving legal proceedings, commercial disputes, and other related digital crime cases. Therefore, the extraction of evidence related to instant meeting software is an important research direction in the current field of digital forensics (*Iqbal et al., 2022*).

Corresponding author
Binglong Li, lbl2017@163.com

Though traditional disk forensics methods have certain data recovery capabilities, their limitations gradually appear in dealing with dynamic data generated by instant meeting software (Kirmani & Banday, 2024). Using disk forensics, forensics personnel can acquire only the files downloaded and saved during a meeting. However, due to the instant data transmission and encryption of the instant meeting software, it is impossible to use disks for forensics. Meanwhile, the running of any process needs to pass through the memory, so instant information such as meeting records will always leave traces in the memory, which requires forensic personnel to turn to a more dynamic memory forensic technology (Zhang et al., 2024). As a result, it become significant to apply memory forensics as a technical means in instant meeting software forensics.

Currently, many scholars have conducted forensic research on instant meeting software. For instance, Barradas et al. (2019) demonstrated the potential of extracting Web client communication records from memory. Van Baar, Alink & Van Ballegooij (2008) investigated the forensic analysis technology of memory-mapped files. The research work of Schatz & Cohen (2017) mentions that memory forensics tools such as Volatility and Rekall can handle general-purpose operating system memory dumps well; Nicoletti & Bernaschi (2019, 2021), Fernández-Álvarez & Rodríguez (2022) have conducted in-memory forensic analysis on other software with instant messaging capabilities such as Telegram Desktop, Skype for Business, and Microsoft Teams, but the research methods are not universal, and each has its own characteristics.

Tencent Meeting, an instant meeting software owned by Tencent, is widely used and active and has more than 400 million users (Kalikiri et al., 2024; Xiao, 2024). Google Meeting (Iqbal et al., 2022), which is similar to Tencent Meeting, has made certain progress in its forensics work, but no research has been conducted on the forensic analysis of Tencent Meeting. Therefore, this article takes Tencent Meeting as the research object to conduct in-depth memory forensic analysis, and aims to extract relevant evidence generated during the running of Tencent Meeting through memory forensic technology.

To solve the problem of Tencent Meeting forensics in memory, this article proposes a method for Tencent Meeting forensics based on memory reverse analysis. The main contributions of this article are as follows:

- The memory of Tencent Meeting is analyzed in detail, and a method and framework of reverse analysis forensics is proposed;
- The memory file of the Tencent Meeting process is analyzed, and a reverse extraction algorithm of Tencent Meeting metadata is developed;
- The different types of data structures in memory are analyzed, the extraction of data streams in transmission is investigated, and a Tencent Meeting data stream engraving algorithm is devised.

## RELATED WORK

### State of the art

Instant meeting software forensics is an important research direction in the field of digital forensics, and its popularity has been increasing in recent years. Nevertheless, due to the

immediacy of such software, the real-time data generated by meetings will not be stored on the disk. Since the operation of any process needs to pass through the memory, instant information such as meeting records always leaves traces in the memory, making it necessary to extract relevant evidence from the memory (*Kirmani & Banday, 2024*). In this context, the memory forensics problem of Tencent Meeting, a widely used instant meeting software in China, has become a research topic worthy of in-depth investigation.

At present, the development of memory forensics technology has attracted much attention from many researchers. The study conducted by *Schatz & Cohen (2017)* not only demonstrates the centrality of in-memory forensics in digital investigations but also promotes the adoption of open-source tools such as Volatility and Rekall. These tools play a crucial role in dealing with memory dumps from Windows operating systems, laying the foundation for subsequent application-specific memory analysis. *Van Baar, Alink & Van Ballegooij (2008)* developed a method for recovering mapped files in memory and discussed the correlation of these files in forensics, which has great significance for understanding the underlying data structures in memory. *Law et al. (2010)* presented an index data structure for analyzing pages in multiple memory dumps to identify static and dynamic pages, providing an important idea for understanding memory structures and improving the reliability of forensic analysis.

In the field of digital forensics, especially in-memory forensics for real-time meetings, different application types and running environments pose challenges to the selection of forensics methods (*Hilgert et al., 2023*; *Ottmann, Breitinger & Freiling, 2024*). The application research of Telegram Desktop provides valuable cases for the memory analysis of various applications. By using the three-stage forensic analysis method, researchers extracted sensitive data from the Windows system, including user account information, communication records, and contacts (*Fernández-Álvarez & Rodríguez, 2022*). *Khalid et al. (2022)* analyzed the memory, disk space and network forensics of Cisco WebEx, and based on the analysis, the researchers extracted user account information, communication records, passwords, *etc.* (*Chang et al., 2013*). *Nicoletti & Bernaschi (2019, 2021)* investigated forensic analysis of Skype for Business and Microsoft Teams to extract information related to VoIP codecs and protocols; *Mahr et al. (2021)* conducted an in-depth disk space forensic analysis of Zoom. They investigated the client-side database in Zoom's data catalog to extract trace information such as contacts, chat history, email addresses, passwords, caches, and user/device configurations. Forensic analysis of instant meeting software, such as Zoom, Microsoft Teams and GoogleMeet, does not directly involve Tencent Meeting, but the experience in extracting meeting records and communication data provides a reference for the memory forensics of Tencent Meeting (*Mahr et al., 2021*; *Ghafarian & Keskin, 2022*; *Iqbal et al., 2022*). The research work suggests that even in applications designed for privacy protection, it is possible to extract valuable information from memory. Also, it indicates that even if user data is deleted after the session ends, it can be still recovered from memory. Although these studies are not directly targeted at Tencent Meeting, their methodology and tool development provide a reference for the memory analysis of Tencent Meeting.

Though certain progress has been made in the field of memory forensics, there is no report on memory reverse analysis forensics for Tencent Meeting. Regarding the development of memory reverse analysis tools, there is currently a lack of tools specifically developed for Tencent Meeting. Meanwhile, no research has been conducted on the extraction and analysis of sensitive information such as data structures, user information and communication records in the memory of Tencent Meeting. As a result, this article proposes a method for Tencent Meeting forensics based on memory reverse analysis, aiming to obtain relevant evidence during Tencent Meeting through memory file reverse analysis.

## Tencent Meeting

Tencent Meeting is a cloud-based video conferencing product that exploits Tencent's extensive experience in audio and video communication. It provides high-definition, smooth, and reliable performance, making it easy to use and suitable for various meeting scenarios. With Tencent Meeting, users can conduct remote audio and video conferences, collaborate online, manage and control meetings, record sessions, send specific invitations, and arrange layouts. For offline office environments, Tencent Meeting provides a meeting room connector that allows existing meeting room equipment to connect to cloud meetings. Additionally, Tencent Meeting has introduced a new generation of intelligent collaboration meeting rooms called Tencent Meeting Rooms, providing enterprises with more advanced and intelligent meeting room solutions.

Figure 1 illustrates the user interface of Tencent Meeting along with its various functional options:

- **Join Meeting**: By clicking "Join" and entering the meeting ID and your nickname, you can enter the specified meeting.
- **Quick Meeting**: Also known as an instant meeting, this feature allows users to initiate a meeting immediately.
- **Schedule Meeting**: This option enables users to schedule a formal meeting in the future by filling in the relevant information.
- **Meeting List**: The meeting list displays upcoming meetings that are either scheduled or favorited by the user, as well as meetings to which the user has been invited (only scheduled meetings are shown; quick meetings do not appear in the list).
- **Historical Meetings**: This section archives content generated from past meetings.

However, once a meeting concludes, neither the meeting records nor the messages exchanged during the communication are stored on disk. These messages can only be viewed by participants in the meeting or call, and they disappear once the meeting or call ends. Therefore, memory forensics is required for data retrieval.

Additionally, when searching for Tencent Meeting on the international internet, VooV Meeting appears. VooV Meeting is the international version of Tencent Meeting, offering similar and interoperable features. Due to the author's lack of an international phone number, this study focuses on the Chinese version of Tencent Meeting.

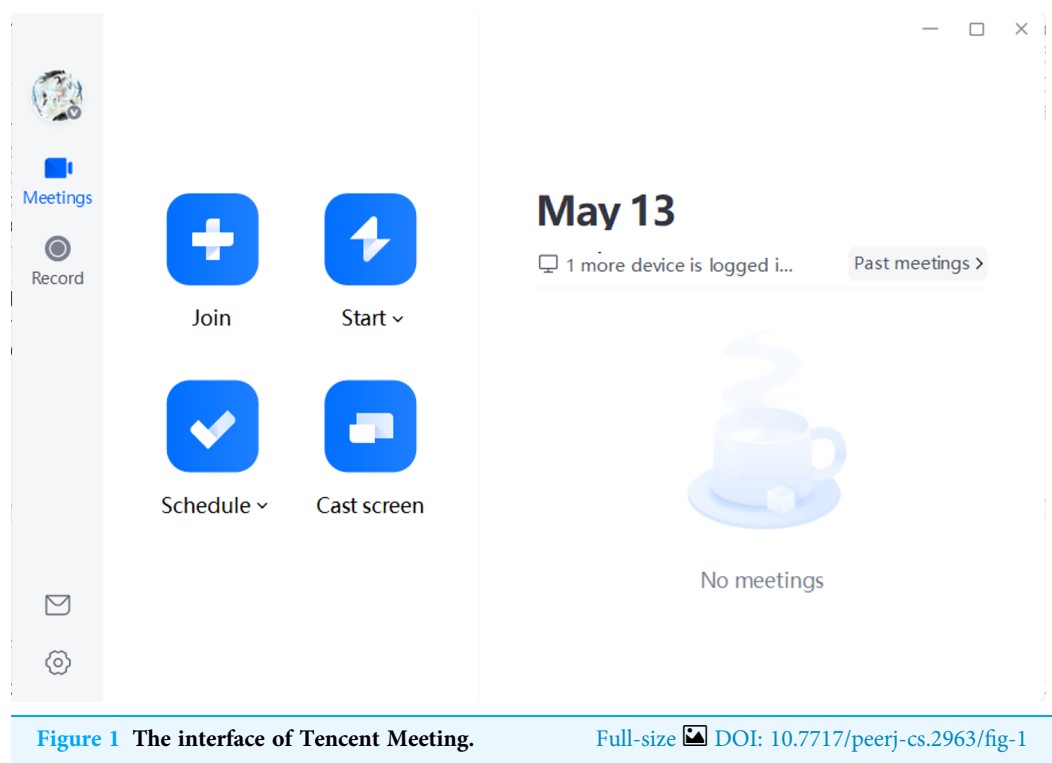

**Figure 1  The interface of Tencent Meeting.**     

## TENCENT MEETING FORENSICS METHOD BASED ON MEMORY REVERSE ANALYSIS

During the operation of Tencent Meeting software, important behavioral trace evidence of users will be left in the memory. This article proposes a method for extracting the evidence of Tencent Meeting behaviors based on memory reverse analysis. The method framework is illustrated in Fig. 2. First, the memory image running Tencent Meeting software is obtained Then, the analysis is divided into two aspects. In one aspect, the memory image is analyzed using the reverse extraction algorithm designed to extract the metadata of Tencent Meeting such as the meeting theme and meeting number. In the other aspect, the designed Tencent Meeting data stream engraving algorithm is employed to analyze the memory image and extract the data stream information generated during the Tencent Meeting attendance. Specifically, the meeting data flow information contains valuable social evidence for forensic investigation such as participants, user identification, and time stamps, as well as behavioral evidence such as message records and documents sent by users during the operation of Tencent Meeting.

### Tencent Meeting metadata reverse extraction algorithm

Tencent Meeting metadata mainly refers to meeting information such as the meeting number and meeting theme. The reverse extraction algorithm of Tencent Meeting metadata mainly involves three steps: the identification of the Tencent Meeting process, the memory extraction of the Tencent Meeting process, and the extraction of Tencent Meeting metadata. The detailed process is illustrated in Fig. 3.

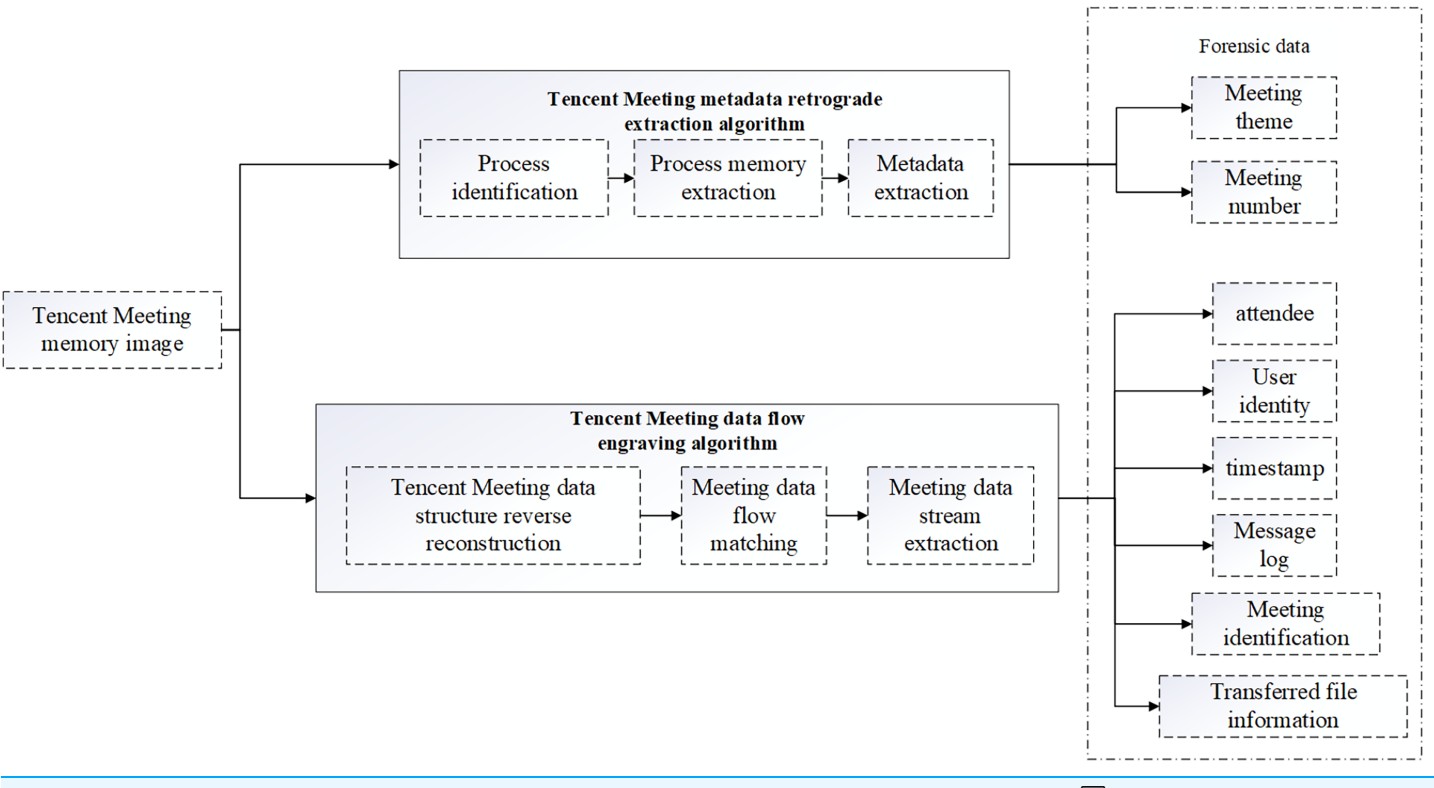

**Figure 2 Tencent Meeting forensics framework based on memory reverse analysis.**

- Create a Windows 10 virtual machine using VMware, create a Tencent Meeting with the theme of "Memory Forensic", and obtain the meeting number 788-357-045.
- Capture the memory image file Windows 10 x64-da1a05fb.vmem on the virtual machine during a meeting.
- Employing Volatility to analyze the memory image file, and use the pslist or psscan plugin to list the process related to Tencent Meeting.
- Extract the process number corresponding to the Tencent Meeting process, namely Process ID (PID) and its virtual address number, as shown in Fig. 4. It can be seen from the figure that in the task manager of the computer, Tencent Meeting runs in the memory under the process name "wemeetapp.exe".
- Use the dumpfiles command of Volatility to dump the files in memory *via* the PID and then transfer them to the findstr command using a pipeline to filter the memory files related to the Tencent Meeting as shown in Fig. 5.
- This section introduces how to analyze the dumped memory files. There are many types of files in the dumped files, including data files (dat), virtual address control block files (vacb), and disk image files (img). The dumped files were opened one by one for analysis. It was found that the time, theme and meeting number of the Tencent Meeting were included in the dat file. The content illustrated in Fig. 6 is consistent with the actual meeting creation, which proves that the required Tencent Meeting metadata information is stored in the dat dump file.

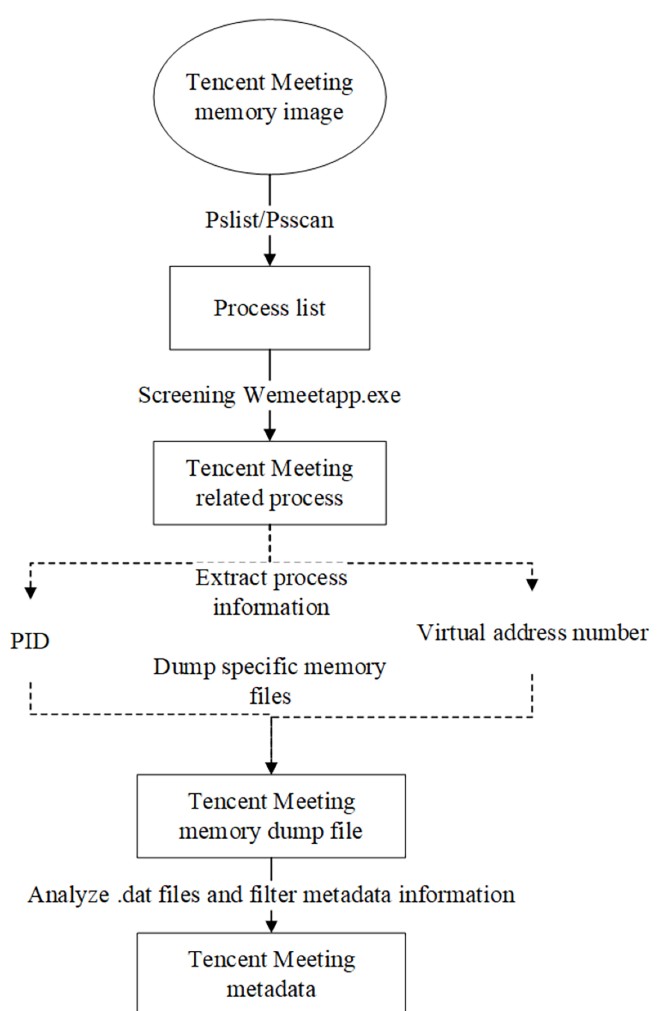

**Figure 3** **Tencent Meeting metadata extraction sub-algorithm.**

Therefore, this article proposes a reverse extraction algorithm for Tencent Meeting metadata. Based on Volatility dumped memory files, the dat files in the dump folder are scanned and then converted into strings in the format of "2024-04-24 08: 43: 31 MemoryForensic 788-357-045", as shown in the above figure. Then, the data is matched to finally obtain the Tencent Meeting metadata.

### Tencent Meeting data flow engraving algorithm

The meeting data flow of Tencent Meeting usually refers to the various data and information flows generated, transmitted and stored in the meeting process, including the basic information of the participating users, the text and file messages sent, as well as the sending time. Some studies have been conducted studies on memory forensics and memory structure analysis (*Zhang et al., 2023*, *2024*). This article proposes a meeting data flow engraving algorithm to obtain meeting data flow information. The proposed algorithm mainly involves three steps: reverse reconstruction of the Tencent Meeting data

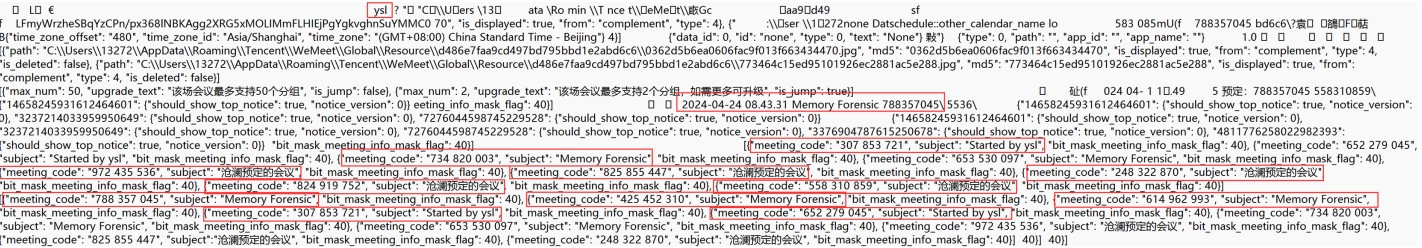

| 8900 | 652 | svchost.exe | 0x9d8a0ed26080 | 19 | – | 0 | False | 2024-04-24 00:39:11.000000 | | N/A | Disabled |
| 8564 | 652 | svchost.exe | 0x9d8a0ecbb080 | 8 | – | 0 | False | 2024-04-24 00:39:14.000000 | | N/A | Disabled |
| 696 | 5196 | wemeetapp.exe | 0x9d8a0e3a50c0 | 12 | – | 1 | True | 2024-04-24 00:39:23.000000 | | N/A | Disabled |
| 3696 | 696 | wemeetapp.exe | 0x9d8a0dbe3080 | 99 | – | 1 | True | 2024-04-24 00:39:25.000000 | | N/A | Disabled |
| 1268 | 3696 | wemeetcrashhan | 0x9d8a0ef07080 | 8 | – | 1 | True | 2024-04-24 00:39:25.000000 | | N/A | Disabled |
| 3716 | 652 | SgrmBroker.exe | 0x9d8a0e3ae080 | 7 | – | 0 | False | 2024-04-24 00:39:28.000000 | | N/A | Disabled |
| 8792 | 652 | svchost.exe | 0x9d8a10010080 | 5 | – | 0 | False | 2024-04-24 00:39:34.000000 | | N/A | Disabled |
| 2640 | 3696 | wmexternal.exe | 0x9d8a08114080 | 16 | – | 1 | True | 2024-04-24 00:39:44.000000 | | N/A | Disabled |
| 8656 | 652 | svchost.exe | 0x9d8a1001c080 | 8 | – | 0 | False | 2024-04-24 00:39:47.000000 | | N/A | Disabled |
| 4944 | 696 | wemeetapp.exe | 0x9d8a0cdf0080 | 11 | – | 1 | True | 2024-04-24 00:39:53.000000 | | N/A | Disabled |
| 692 | 3696 | wemeetapp.exe | 0x9d8a0ef29080 | 10 | – | 1 | True | 2024-04-24 00:40:11.000000 | | N/A | Disabled |
| 3076 | 3696 | wemeetapp.exe | 0x9d8a0ec91340 | 6 | – | 1 | True | 2024-04-24 00:40:15.000000 | | N/A | Disabled |
| 2988 | 3696 | wemeetapp.exe | 0x9d8a0dab1080 | 14 | – | 1 | True | 2024-04-24 00:40:15.000000 | | N/A | Disabled |
| 5164 | 3696 | wemeetapp.exe | 0x9d8a0e457080 | 13 | – | 1 | True | 2024-04-24 00:40:21.000000 | | N/A | Disabled |
| 6972 | 652 | svchost.exe | 0x9d8a0de962c0 | 4 | – | 0 | False | 2024-04-24 00:40:22.000000 | | N/A | Disabled |
| 7736 | 652 | svchost.exe | 0x9d8a0ee6c080 | 3 | – | 0 | False | 2024-04-24 00:41:31.000000 | | N/A | Disabled |
| 7496 | 652 | MpDefenderCore | 0x9d8a0da31080 | 11 | – | 0 | False | 2024-04-24 00:41:33.000000 | | N/A | Disabled |
| 3756 | 776 | dllhost.exe | 0x9d8a0edee2c0 | 6 | – | 0 | False | 2024-04-24 00:41:35.000000 | | N/A | Disabled |
| 9020 | 1740 | audiodg.exe | 0x9d8a0e3ad080 | 7 | – | 0 | False | 2024-04-24 00:41:37.000000 | | N/A | Disabled |
| 1012 | 3696 | wemeetapp.exe | 0x9d8a0ef0c080 | 9 | – | 1 | True | 2024-04-24 00:43:03.000000 | | N/A | Disabled |
| 1888 | 3696 | wemeetapp.exe | 0x9d8a0ecbc080 | 9 | – | 1 | True | 2024-04-24 00:43:27.000000 | | N/A | Disabled |
| 2264 | 5196 | notepad.exe | 0x9d8a0ed85080 | 3 | – | 1 | False | 2024-04-24 00:44:00.000000 | | N/A | Disabled |
| 8816 | 5100 | SearchProtocol | 0x9d8a0eeb9080 | 6 | – | 1 | False | 2024-04-24 00:44:01.000000 | | N/A | Disabled |
| 4668 | 5100 | SearchFilterHo | 0x9d8a112f1340 | 3 | – | 0 | False | 2024-04-24 00:44:01.000000 | | N/A | Disabled |
| 7936 | 652 | svchost.exe | 0x9d8a1150b340 | 7 | – | 0 | False | 2024-04-24 02:16:15.000000 | | N/A | Disabled |
| 2144 | 652 | TrustedInstall | 0x9d8a1153f340 | 9 | – | 0 | False | 2024-04-24 02:16:15.000000 | | N/A | Disabled |
| 8716 | 776 | TiWorker.exe | 0x9d8a10d8d340 | 8 | – | 0 | False | 2024-04-24 02:16:17.000000 | | N/A | Disabled |
| 9000 | 776 | backgroundTask | 0x9d8a0ec78080 | 13 | – | 1 | False | 2024-04-24 02:16:19.000000 | | N/A | Disabled |
| 5996 | 776 | RuntimeBroker. | 0x9d8a0edb4080 | 7 | – | 1 | False | 2024-04-24 02:16:27.000000 | | N/A | Disabled |
| 8336 | 2672 | cmd.exe | 0x9d8a0edc2080 | 0 | – | 0 | False | 2024-04-24 02:17:13.000000 | 2024-04-24 02:17:14.000000 | | Disabled |
| 1672 | 8336 | conhost.exe | 0x9d8a0ed2a080 | 0 | – | 0 | False | 2024-04-24 02:17:13.000000 | 2024-04-24 02:17:14.000000 | | Disabled |

**Figure 4** System process extraction. 

```
PS F:\volatility3-develop> python vol.py -f "F:\Virtual Machines\
Win10\Windows 10 x64-da1a05fb.vmem" windows.dumpfiles --pid "3639
" | findstr "wemeetapp.exe"
```

**Figure 5** Extract the system process. 

**Figure 6** Memory dump file extraction information. 

structure, meeting data flow matching, and meeting data flow extraction. The design process of each stage is as follows:

(1) Reverse reconstruction of the Tencent Meeting data structure

The main purpose of the reverse reconstruction of the Tencent Meeting data structure is to build the structural information of Tencent Meeting to store the data of the meeting in memory. The basic design ideas are described below:

- Employ VMware to set up Win10 virtual machines, create and participate in Tencent Meeting, and send text messages such as "how are you" and file messages such as "story for us.mp3".

- Capture the memory image file Windows 10 x64-da1a05fb.vmem on the virtual machine.

- Manually open the obtained memory image file "Windows 10 x64-da1a05fb.vmem" using the WinHex hexadecimal editor, and reverse engineer the data structures of the text and file messages by analyzing its hexadecimal data.

- Perform reverse reconstruction of the text message structure: Search the sent text message "how are you", and the search result is demonstrated in Fig. 7. By analyzing the hexadecimal data of the memory image using WinHex, the messages transmitted in JSON format during the meeting can be reverse-engineered. "how are you" and "text_elem_content" form a key-value pair, where "how are you" is stored in "text_elem_content". Focusing on this key-value pair, forward analysis reveals a preceding key-value pair "elem_type":0. After verification by multiple sets of data, although this "elem_type" has no forensic significance, it acts as verification data within the same object as the message key-value pair. Continuing with the forward analysis, "{" and "[" can be seen in the graph, indicating that some data may exist in the same object or array. Continuing with the backward analysis, "}" can be found, indicating that the above two groups of key-value pairs exist in the same object. Since there are multiple groups of objects, several groups of key-value pairs associated with the message can be found, such as \"nickname\": \"ysl\", \"unique_id\": \"144115391065829678\", *etc.*, and this agrees with the actual situation. Finally, "]" is found, proving that multiple groups of objects are stored in the array" message_elem_array", *i.e.*, in the graph stored in the same "[]". Therefore, the array as a whole represents all the relevant information of the message, and according to its integrity, the validity of this part of the data can be determined.

- When searching for "how are you", multiple search results can be obtained, and the validity of the search result can be determined by analyzing whether it has verification data, so the final result is obtained after filtering, as shown in Fig. 7.

- By analyzing the overall data in the above figure, it can be seen that most of the objects exist as necessary components in an array, without forensic significance, and they are only used for verification. Verification can prove that this part is indeed a real communication record rather than a forged one. Some objects within it can also help identify the record as belonging to a specific communication, preventing messages related to other meetings from confusing the results. Therefore, from the perspective of forensic analysis of other information, the text data structure of Tencent Meeting in memory with forensic significance is finally obtained and shown in Fig. 8.

- Reversely reconstruct the file message structure: Search the sent MP3 file "story for us" in the memory image file "Windows 10 x64-da1a05fb.vmem", and the search result is presented in Fig. 9. It can be seen from the figure that the file names "story for us" and "name" form a key-value pair and are stored in the object "file_info" together with other sets of key-value pairs. Through the analysis of the surrounding data centered on the key-value pair, several groups of key-value pairs associated with the file can be found,

Figure 7 "How are you" message search results.

such as \"nickname\": \"ysl\",\"size\": 5368895,\"report_file_type\": \"mp3\", and this is consistent with the actual situation. Multiple objects are also stored in the array" message_elem_array", so the array as a whole represents all relevant information about

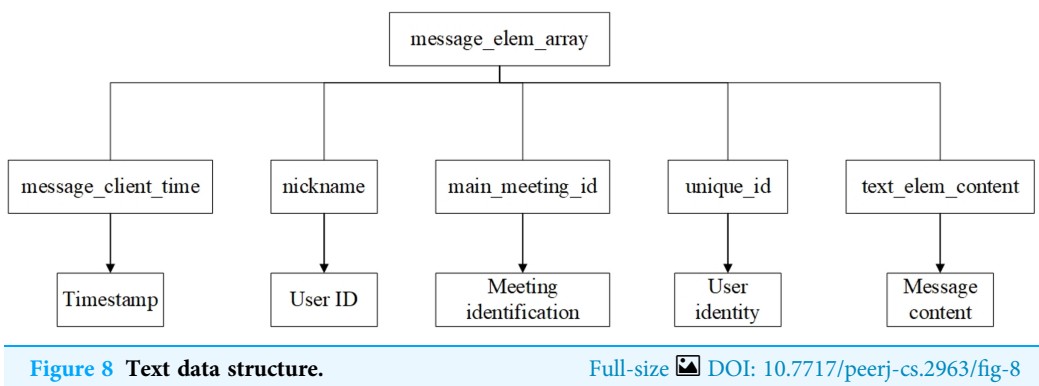

**Figure 8  Text data structure.**               

**Figure 9  "Story for us" message search results.**               

the file, which is the same as the text, and through backward analysis, the array end position can be found, forming a closed loop. The validity of this part of the data can be also determined by the integrity of the array, preventing multiple invalid files from appearing during scanning.

Through the analysis of the overall data in the figure, it can be seen that most of the objects are a necessary component of an array, with no forensic significance, and they can only be used for verification. Therefore, from the perspective of forensics analysis, the finally obtained file data structure of Tencent Meeting in the memory with forensic significance is demonstrated in Fig. 10.

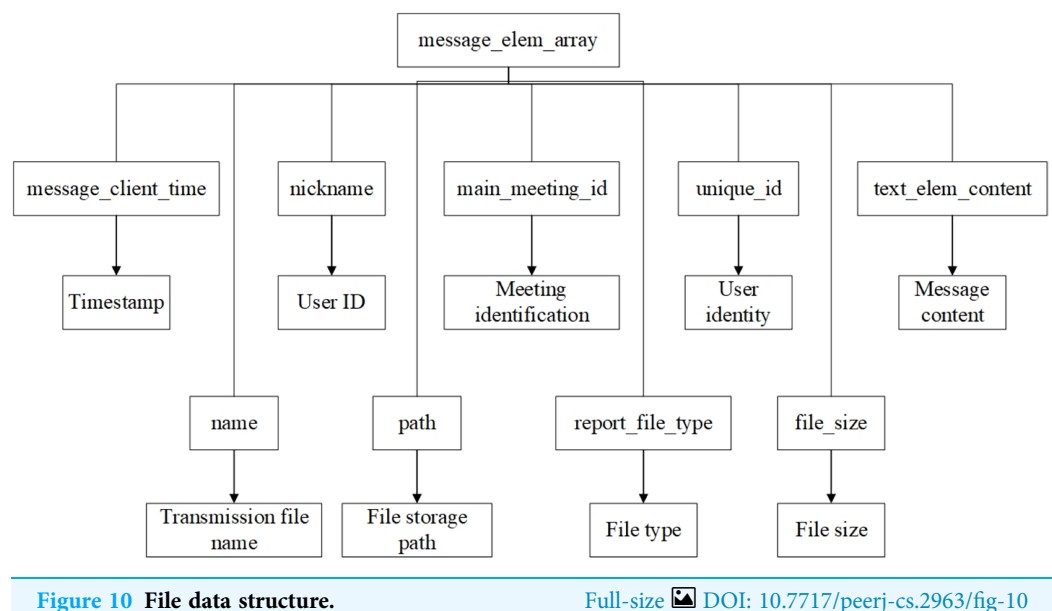

**Figure 10 File data structure.**

(2) Meeting data flow matching

Meeting data flow matching is devised to accurately identify and locate the data structures of Tencent Meeting from memory image files. To enhance search efficiency and reduce mismatching, this article adopts the Knuth-Morris-Pratt (KMP) algorithm and optimizes it to adapt to the characteristics of the Tencent Meeting data structure. The specific details are as follows (*Lu, 2019*):

According to the reconstructed data structure, message_elem_array is a core data structure. Other key information, such as message timestamp, user ID, meeting ID, and message content, have a close correlation to message_elem_array and are stored in this array. This intrinsic data correlation allows the algorithm to locate message_elem_array first to narrow the scope of subsequent searches, thereby achieving higher search efficiency.

When message_elem_array is matched, a second round of matching is conducted, *i.e.*, matching the required key-value pairs according to the data structure of the text message and file message in memory. Several pieces of message_elem_array data will be matched in the first round of search. Therefore, in the second round, it needs to use validation data such as key values to filter "elem_type":0 to exclude invalid message_elem_array data.

Using the phased string matching strategy, the algorithm can effectively match the data structure of Tencent Meeting forensics from the memory image file. By narrowing the search space and optimizing the search scope, this strategy enhances search efficiency and matching accuracy, and it has important practical application value for data extraction in memory forensics.

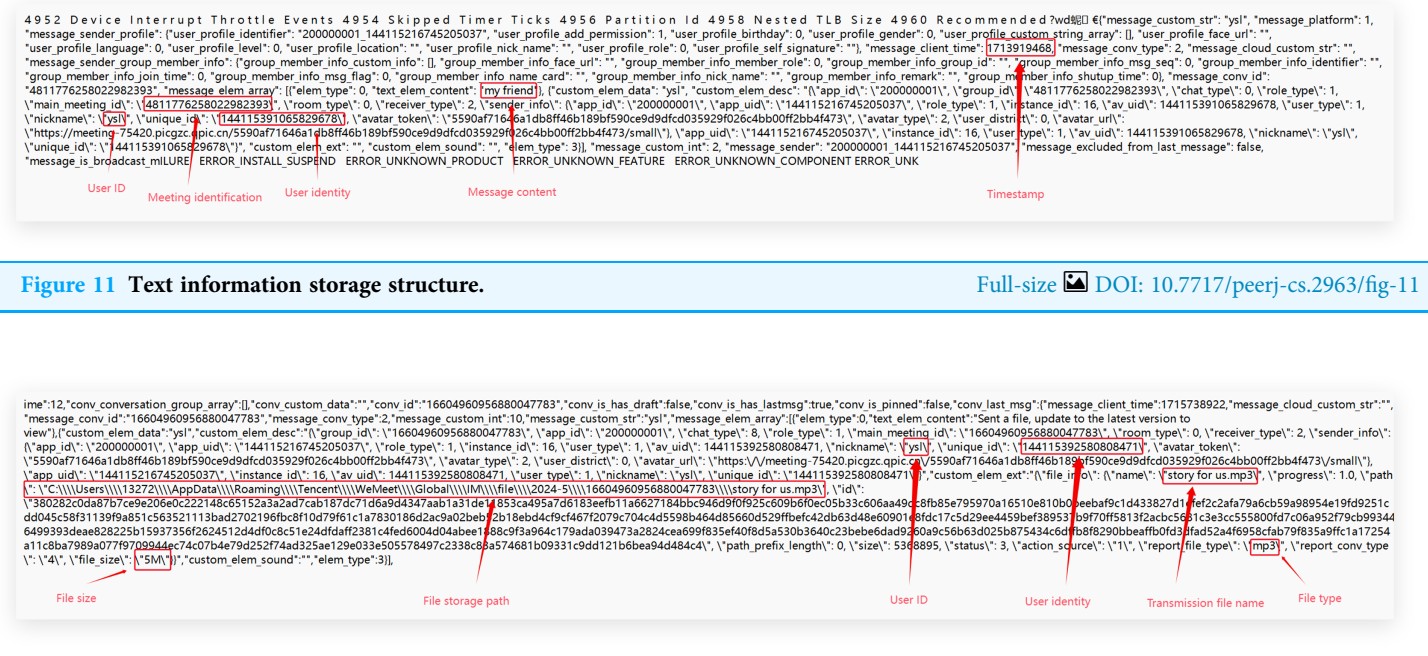

**Figure 11 Text information storage structure.**

**Figure 12 File information storage structure.**

(3) Meeting data stream extraction

The Meeting data stream extraction stage aims to extract specific meeting data stream information precisely from the identified memory region, so a carving algorithm is proposed in this article as follows:

- Use the matching position of message_elem_array and other key data structure key-value pairs to determine the specific extraction range of each identifier.
- Based on the matching location of message_elem_array, extract the data blocks of message_client_time, nickname, unique_id, and text_elem_content.
- Analyze the message_elem_array data block to identify the start and end locations of each message. Then, for each message, extract and parse the timestamp, the nicknames and unique identifiers of the sender and receiver, and the specific message content.
- Construct a complete message object including message elements, time stamps, user information and message content, and integrate all message objects in chronological order to form the meeting process, as illustrated in Figs. 11 and 12.
- Organize the integrated meeting process into a data flow to ensure the logical sequence and data integrity, and mark key events in the data flow, such as the start of the meeting, file transfer, and important messages.
- Collate the data into formatted report outputs including key events and complete meeting flow.

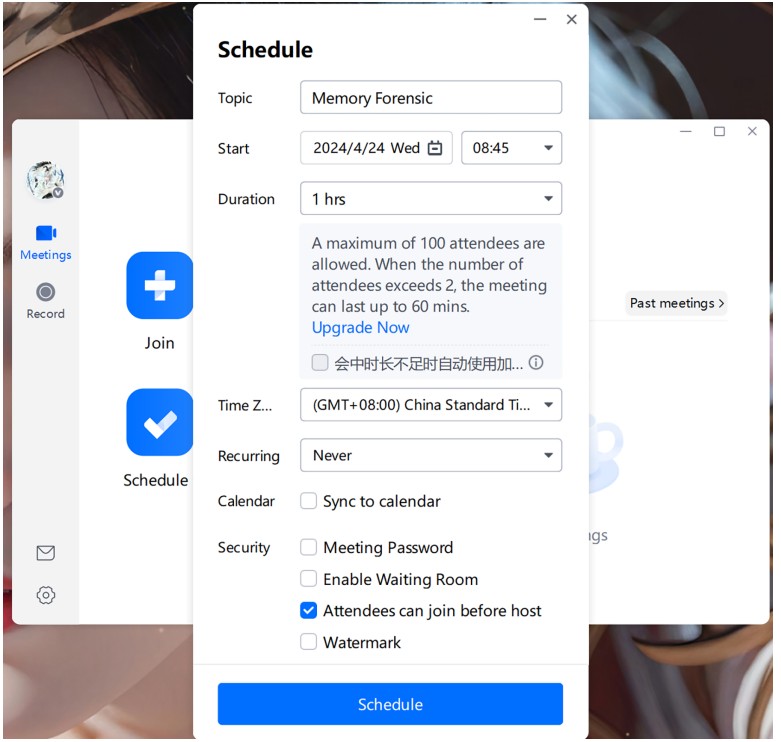

**Figure 13  Schedule meeting interface.**

After the above steps, the meeting data stream engraving algorithm can accurately extract and reconstruct the meeting data, providing a basis for further analysis and forensics.

# EXPERIMENT AND DISCUSSION

## Experimental configuration

Experimental software: Tencent Meeting 3.31.2.441

Operating system: Windows 10 Professional

Experimental tool: Vmware16, Volatility3, WinHex20

First, create a Windows 10 virtual machine with VMware, and then install and start Tencent Meetings.

When opening Tencent Meeting, we can see options such as "Start", "Join","Schedule", and "Cast screen". The goal of this experiment is to find information about a specific meeting, including its theme, participants, and meeting content, through the method proposed in this article, so as to provide evidence for later criminal forensics. Therefore, we start and join a meeting with the theme "Memory Forensic" and send messages in the chat box for later verification. Figures 13, 14 and 15 display some scenes during the creation and operation of Tencent Meeting:

We suspend this virtual machine to obtain a memory image file. Then, we locate the .vmem image snapshot file in the virtual machine's path, as shown in Fig. 16, for subsequent analysis of this image snapshot using tools such as Volatility.
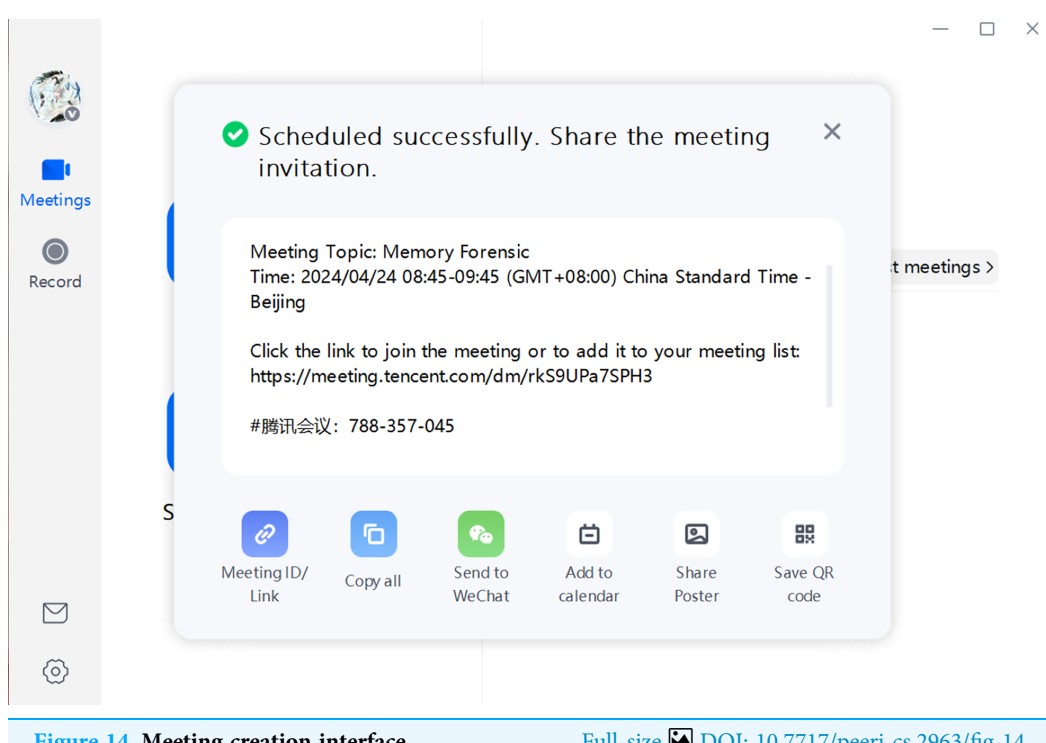

**Figure 14  Meeting creation interface.**

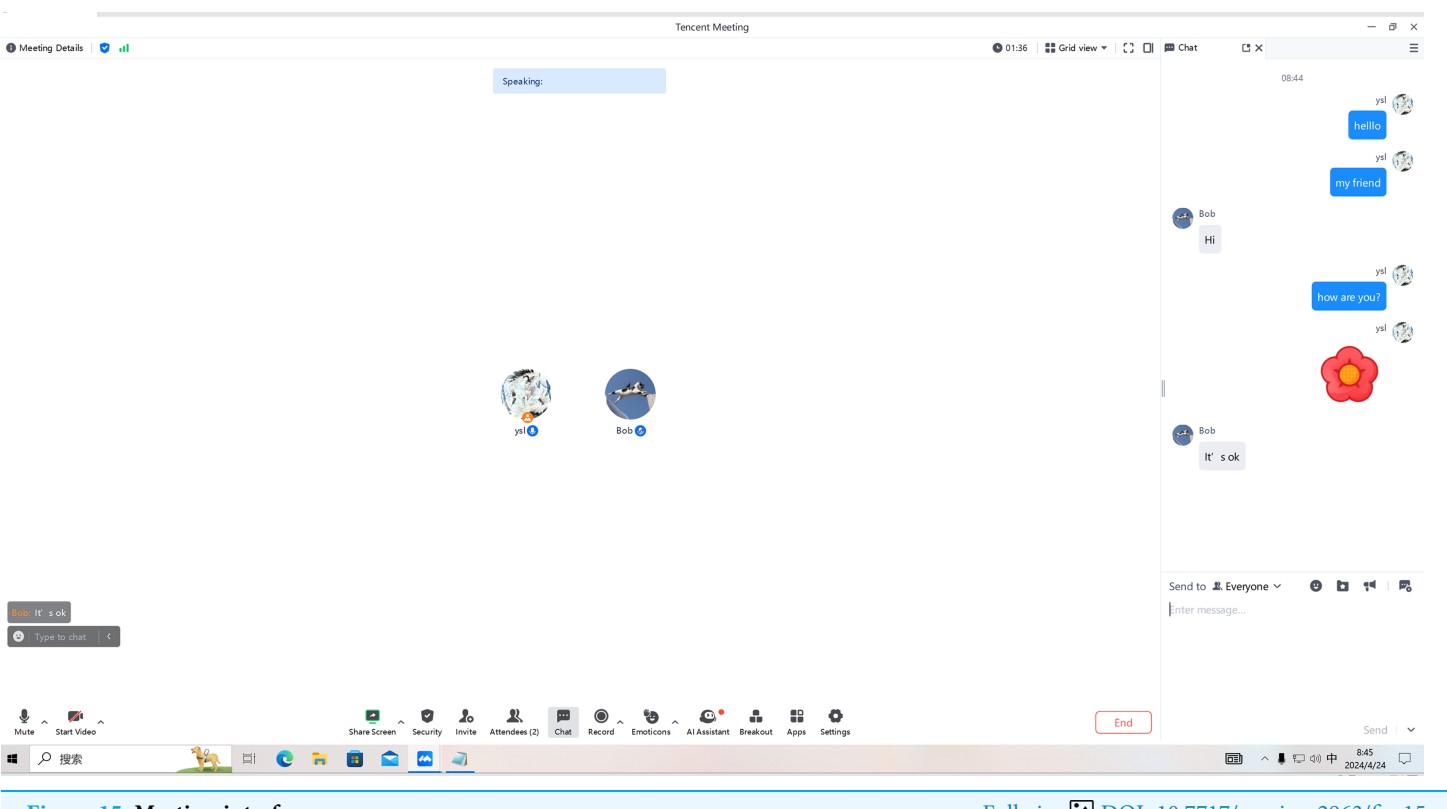

**Figure 15  Meeting interface.**

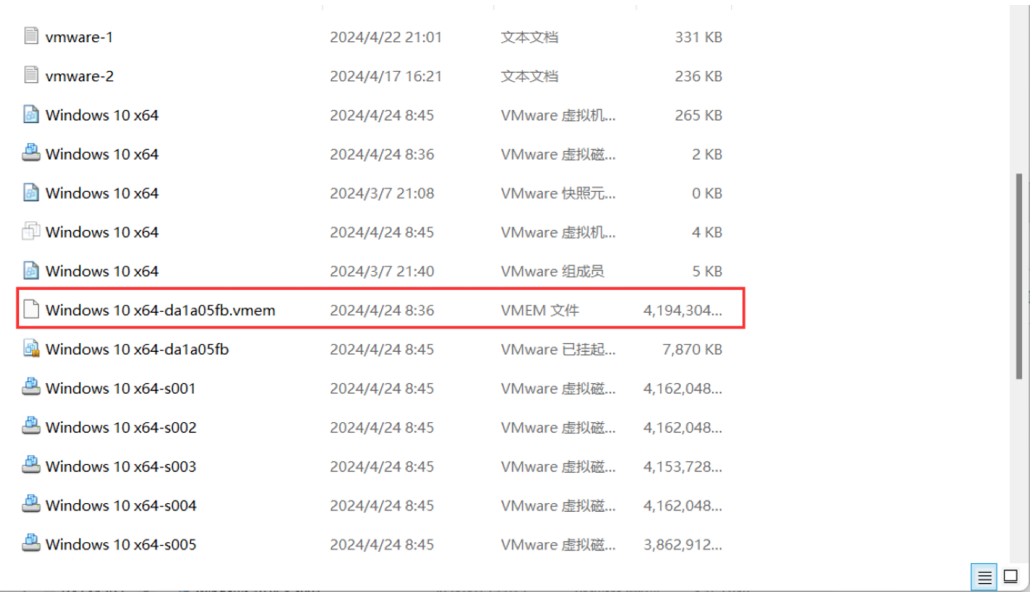

**Figure 16  Memory image location.** 

## Experimental verification

Suppose that Tencent Meeting was used to plan and execute a sophisticated online fraud case, where fraudsters used the instant messaging function of Tencent Meeting to deceive victims to invest through false identities and fake backgrounds. To conceal their criminal behavior, the fraudsters also used Tencent Meeting's document transfer function to transmit forged investment reports and contract documents.

After the case was revealed, forensic analysts found that the data stored in Tencent's meeting memory could be used as key evidence to uncover the criminal behavior of fraudsters. Due to the large amount of communication and file transfer conducted by fraudsters in Tencent Meeting, the memory may contain their real identity information, criminal plans, personal information of victims, as well as detailed records of fraud activities.

Therefore, forensic analysts need to perform a forensic analysis of the memory of the Tencent Meeting to obtain this key evidence. This will help determine the true identity of fraudsters, expose their criminal activities, and protect more potential victims. By conducting a forensic analysis of Tencent Meeting memory, forensic personnel are expected to reveal the criminal network of fraudsters and take appropriate legal action.

When conducting a forensic analysis of Tencent Meeting memory image, it needs to consider various conditions when Tencent Meeting is running and after it is shut down. In this experiment, the proposed inverse extraction algorithm of Tencent Meeting software metadata was employed to identify and extract the relevant information of the Tencent Meeting process "wemeetapp.exe" by using the pslist/pscan plugin of the Volatility tool.

**Figure 17 Comparison between pslist and psscan.**

**Table 1 Meeting theme and minutes.**

| Time | meeting_code | Subject |
|------|-------------|---------|
| 2024-04-24 08.43.31 | 788 357 045 | Memory forensic |
| | 307 853 721 | Started by ysl |
| | 652 279 045 | Started by ysl |
| | 734 820 003 | Memory forensic |
| | 824 919 752 | Scheduled meeting |
| | 558 310 859 | Scheduled meeting |
| | 614 962 993 | Memory forensic |

After using the pslist/pscan plugin of the Volatility tool respectively to identify and extract the Tencent Meeting process, it can be found that Tencent Meeting running time does not lead to a significant difference, as illustrated in Fig. 4.

After the Tencent Meeting was shut down, by using the pslist/pscan plugin of the Volatility tool to identify and extract the Tencent Meeting process, it can be found that the output results of different commands are inconsistent. The comparison results are shown in Fig. 17.

After the process information was obtained, files were dumped by the process ID. After the dat file of the Tencent Meeting was dumped by PID, the theme of the past meeting was extracted from the series of files output by the Tencent Meeting process in combination with the metadata in a specific format, and the start time of the latest meeting was recorded and determined, as listed in Table 1.

After the meeting number and the topic of the meeting are obtained, it is also necessary to obtain the meeting data to provide strong evidence of digital crime. Therefore, the user behavior data was carved by the proposed meeting data stream engraving algorithm. Previous studies have known the data structure of Tencent Meeting to store various information. After the data source file was obtained, the designed engraving algorithm was employed to scan, match and extract the entire memory image file, and the user behavior data was output, as shown in Fig. 18 and Table 2.

**Figure 18 String search results.**

**Table 2 User meeting data flow-message information.**

| message_client_time | main_meeting_id | Nickname | unique_id | text_elem_content |
|---|---|---|---|---|
| 1713919468 | 4811776258022982393 | ysl | 144115391065829678 | my friend |
| 1713919508 | 4811776258022982393 | Bob | 144115392378737712 | [fireworks] |
| 1713919480 | 4811776258022982393 | ysl | 144115391065829678 | how are you? |
| 1713919503 | 4811776258022982393 | Bob | 144115392378737712 | It's ok |
| 1713919462 | 4811776258022982393 | ysl | 144115391065829678 | helllo |

**Table 3 User meeting data flow-file information.**

| Nickname | unique_id | Name | report_file_type | file_size |
|---|---|---|---|---|
| ysl | 144115391065829678 | story for us.mp3 | mp3 | 5 M |
| ysl | 144115391065829678 | test.txt | txt | 1 K |
| ysl | 144115391065829678 | analysis515.docx | docx | 1 M |

According to the output, the same meeting identifier can confirm that the message comes from the same meeting. Even if each user has the same ID, each has his/her own user ID and can still be distinguished. The timestamp indicates the time when the message was sent. The time converted to Beijing time is consistent with the fact. The content of the message can provide important support for evidence acquisition.

For file transmission, the experiment also utilized the meeting data stream engraving algorithm to extract the user's behavior data of file transmission, and the results are shown in Table 3.

**Table 4 Comparison of methods.**

| Comparison of results | Meeting number | Meeting theme | Times-tamp | Participating user | Text message | File message | File type | File size |
|---|---|---|---|---|---|---|---|---|
| Google Meeting memory evidence extraction method | √ | √ | √ | √ | √ | × | × | × |
| Tencent Meeting forensics method based on memory reverse analysis | √ | √ | √ | √ | √ | √ | √ | √ |

By analyzing the output results, it can be confirmed that the content of the transmitted file is consistent with reality. In the experiment, the information communication and transmission files in the meeting were recovered through the meeting data stream engraving algorithm, which played a crucial role in obtaining the real identity information of the criminals, the crime plan, the personal information of the victims, as well as the detailed records of the fraud activities, and it also provided key evidence for the punishment of the crime.

The above experiments can demonstrate the feasibility and effectiveness of our developed Tencent Meeting forensics method based on memory reverse analysis. It not only regenerates the data transmission of Tencent Meeting but also has great significance for the forensics of instant meetings.

## DISCUSSION

In this article, our proposed Tencent Meeting forensics method based on memory reverse analysis directly reverse-analyzes the whole memory file and extracts the meeting data including text and file messages from memory through process memory extraction, data structure reverse reconstruction, and meeting data flow matching. It provides a new perspective and tool for in-memory forensics in instant meeting applications.

In a forensic analysis of Google Meeting, *Iqbal et al. (2022)* presented a memory trace extraction tool to automatically extract evidence *via* unstructured string analysis. Meeting information was obtained by analyzing the memory usage of Google Meeting on browsers such as Chrome, Firefox, and Edge (*Khalid et al., 2022*). Nevertheless, this method requires the browser to obtain memory data, and the file information transmitted during the meeting cannot be acquired.

By comparing the forensics methods of the two meetings, this article also successfully extracted the meeting time, meeting number, topic, participants, message records and transmitted files based on the forensics results of Google Meeting, proving that our method can effectively extract more complete meeting information from the running memory. It provides important digital evidence for digital crime investigation. The comparison results are listed in Table 4.

Through the above comparative analysis, it can be found that the Tencent Meeting memory forensics method proposed in this article demonstrates obvious advantages in terms of methodology, experimental results, performance, research innovation, and practical application value. This method makes an in-depth investigation of memory forensics for Tencent Meeting, a desktop application, and it fills the existing research gap.

Through experimental verification, the proposed method can effectively extract the relevant data of the meeting, showing significant practical application value for legal investigation and crime forensics. Additionally, the research of this article is not only applicable to Tencent Meeting but also can be analogically extended to the memory forensics of other meeting software, so it has wide applicability and promotion prospects.

## CONCLUSIONS

This article investigates the problem of memory forensics of Tencent Meeting and points out that the data of this software is stored in the memory in a temporary format and disappears when the system is shut down or in standby mode. Therefore, it is necessary to conduct memory forensics for the data acquisition of Tencent Meeting. To this end, this article proposes a method of Tencent Meeting forensics based on memory reverse analysis. By using tools such as Volatility and Winhex, meeting metadata reverse extraction algorithm and meeting data stream engraving algorithm, the proposed method can successfully extract meeting information, which provides important support for forensics.

The advantage of our method is that it can restore complete information and user behavior data of Tencent Meeting, providing evidence for digital crime forensics. This article presents a complete set of Tencent Meeting forensics analysis methods, which can be applied to similar software.

However, the extraction of real-time voice data is still a challenge, and it will be the focus of future forensic work. Extracting voice data involves the issue of audio data encoding. In future work, relevant professional knowledge can be exploited to extract audio data from instant meetings. This is of great significance for forensic work and will also be the focus of subsequent forensic investigations.

## ACKNOWLEDGEMENTS

The authors would like to thank all the reviewers who participated in the review and MJEditor for their linguistic assistance during the preparation of this manuscript.

### Funding

This work was supported by the National Natural Science Foundation of China (No.60903220). The funders had no role in study design, data collection and analysis, decision to publish, or preparation of the manuscript.

### Grant Disclosures

The following grant information was disclosed by the authors:
National Natural Science Foundation of China: 60903220.

### Competing Interests

The authors declare that they have no competing interests.

## Author Contributions

- Shilong Yu conceived and designed the experiments, performed the experiments, analyzed the data, performed the computation work, prepared figures and/or tables, authored or reviewed drafts of the article, and approved the final draft.
- Binglong Li performed the computation work, authored or reviewed drafts of the article, and approved the final draft.
- Lin Zhu analyzed the data, prepared figures and/or tables, and approved the final draft.
- Heyu Zhang performed the experiments, authored or reviewed drafts of the article, and approved the final draft.
- Sen Yang performed the computation work, prepared figures and/or tables, and approved the final draft.
- Zhangxiao Li performed the computation work, prepared figures and/or tables, and approved the final draft.
- Wenzheng Feng performed the computation work, prepared figures and/or tables, and approved the final draft.

## Data Availability

The data is available at FigShare: Yu, Shilong (2025). File1.7z. figshare. Dataset. https://doi.org/10.6084/m9.figshare.28481333.v1.

## Supplemental Information

Supplemental information for this article can be found online at http://dx.doi.org/10.7717/peerj-cs.2963#supplemental-information.

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
