# Peer review of "Tencent Meeting forensics based on memory reverse analysis"

_PeerJ Computer Science, doi:10.7717/peerj-cs.2963_

## Round 0.1 · original submission · Major Revisions

Please address the concerns of both reviewers

Reviewer 1 ·

Basic reporting

#0
This research focuses on Tencent Meeting, an instant meeting software, and the lack of existing forensics methods for it. The paper proposes a memory-based forensics approach that involves reverse engineering Tencent Meeting's memory structure. This method aims to extracts crucial meeting metadata (time, number, topic) and data flow information (participants, messages, files), providing valuable digital evidence for investigations.

#1
The whole paper is difficult to follow, and thus it makes for a difficult read. The English is grammatically correct but the content of the paper often lacks important details. For instance, the Experiment section is confuse, and thus hard to read. The study needs a systematic approach, and a better presentation of results. For instance, a table with the most meaningful results would
help readers.

#2
The paper lacks a proper introduction to the Tencent Meeting software as not everybody is familiar with the software (A web search about "Tencent Meeting" returns info about VooV Meeting -- it seems that "Tencent Meeting" is for the Chinese market, while "VooV Meeting" is the international version). Nonetheless, the paper needs to properly introduce the "Tencent Meeting" software. Readers not familiar with "Tencent Meeting" only discover that it is a Windows desktop application on line 109 ("...runs in the memory under the process name "wemeetapp.exe"). Nonetheless, some doubts persist: i) is it a Win32 application or an UWP application? ii) is it built on top of Electron? All of these need to be clarified. Graphically, a screenshot or two of the application would provide some help to readers to have a grasp of what is "Tencent Meeting".

#3
The absence of description/introduction to the Tencent Meeting software is not the only problem of the paper: it lacks important details, making for a difficult read. For instance, in line 141, the paper states: "Use the WinHex tool to open the memory image file 'Windows 10 x64-da1a05fb.vmem' to reverse the data structure of text and file message": The reader is left wondering whether the reverse process is manually performed through WinHex hexadecimal
editor.

#4
(line 65 states that Tencent Meeting is widely used and has more than 400 million users -- are all users from China?) Another question arises: does the findings about "Tencent Meeting" also apply to "VooV Meeting"?

#5
The paper does not provide the following important data: i) the version of the Tencent Meeting software tested; ii) version/subversion of Windows (Windows 10 has many sub-versions); iii) Version of Volatility; iv) Version of other tools (e.g., WinHex).

#6
Line 48:"However, due to the instant data transmission and encryption of the instant meeting software, it is impossible to use disks for forensics." => Why is it impossible to use disks for forensics? Is it because content kept in disk is encrypted or it is due to the fact that nothing is kept in disk?

#7
Line 143 (...): It appears that you are describing a JSON-formatted message. This should be made explicit in the text.

#8
Line 163: "By analyzing the overall data in the above figure, it can be seen that most of the objects exist as necessary components in an array, without forensic significance, and they are only used for verification"
=> Can you elaborate more about the meaning of verification in this context?

Some observations and minor questions:

Abstract - "combines the whole memory file"
=> What do you mean by "whole memory file"?

Line 60: "Can well handle"
=> Revise English

Line 102: "and obtain the meeting number: 788 357 045"
=> Is the meeting number given as an example, just to provide the reader with an idea of the format of a meeting number?
=> What's the importance of the Meeting number?

Line 155: "144115391065829678" - does a unique_id in "Tencent Meeting"
software convey a timestamp? (like for example, Twitter ID messages?)

Line 161: "...the validity of the valid search result can be determined..."
=> Avoid repetition of "valid"

Line 188: Missing bibliographic reference for the "Knuth-Morris-Pratt (KMP)"
algorithm.

Line 244: ..., **it** needs to consider... - replace 'it' with 'one'

Figure 1: there a "t" missing in "Meeing"

Experimental design

The experimental design is hard to follow, and almost impossible to reproduce. A diagram documenting the steps should be provided.
In its current form, a reader cannot reproduce the experiment.

Validity of the findings

The validity of the findings is hard to assess as the paper does not provide, at least in a clear form, its main results. As the experiments and its parameters are hard to decrypt, it is also difficult to properly judge the findings. As stated above, the paper needs to be properly organized. This includes to present the results in a proper manner, possibly resorting to tables, diagrams and some listings (the paper provides some of these, but there are insufficient).

Additional comments

N.A.

Cite this review as

Reviewer 2 ·

Basic reporting

The quality of Figure 4&9 require improvement to ensure they are visually clear, and the label for Figure 5,7 should be better organized and more descriptively detailed to enhance its interpretability and avoid misunderstanding like adding single quotes to the searching string. There is a small typo in Figure 1 engraving meeting data. For Figure 2, it is better to write as Analyze .dat file

Experimental design

Overall pretty good. But in the design, it is better to have steps listed in bullet points, making it easier for the readers to follow and understand.

Validity of the findings

Findings look good and supported with the goals and research data.

Additional comments

The paper didn't have a future work discussion. Authors could consider having what can be also done in the future of this work if applied.
The paper pointed out a good way for general digital forensics when encountered encryption problems.

Cite this review as

---

## Round 0.2 · Minor Revisions

Please address the remaining minor revisions.

Reviewer 1 ·

Basic reporting

#1
The authors have addressed all issues raised in the previous review.

#2
Some observations:

a) Hyphenation of the text appears to be broken (not relevant from the scientific point of view)

b) Line 135: "Tencent Meeting introduction"
=> "Tencent Meeting" (remove introduction from the section title)

c) Line 163: ". Due to the author's lack of an international phone number, this study focuses on the domestic version of Tencent Meeting. "
domestic -> Chinese, to remove any ambiguity to readers

d)Line 164: Nonetheless, the research methods and findings presented in this paper are also applicable to VooV Meeting.

=> Has this been confirmed? Otherwise, it would be prudent to state "We **believe** that the research methods...", or even skip that part as testing it was impossible.

e)
Line 253: "Some of the objects within it can also help identify the record as belonging to a specific communication, preventing messages from other meetings from confusing the results. "
SUGESTION: (to avoid the double "from" and correct "of the" at the beginning of the sentence.)
=> "Some objects within it can also help identify the record as belonging to a specific communication, preventing messages related to other meetings from confusing the results."

Experimental design

OK.

Validity of the findings

OK.

Additional comments

N.A.

Cite this review as

---

## Round 0.3 · Minor Revisions

The authors addressed all the technical concerns from the reviewers. Still, the paper suffers from some grammatical mistakes and typesetting issues. The authors should carefully proofread the paper to improve the writing. In addition, the following typesetting issues should be fixed: make uniform the style of bulleted lists; move figures in text; make lines such as those from 186 to 195 a bulleted or numbered list (many instances of this pattern are present); rewrite paragraphs having (many) nested semicolons such as those at lines from 355 to 362; decrease the line spacing of the references.

**Language Note:** The Academic Editor has identified that the English language must be improved. PeerJ can provide language editing services - please contact us at [email protected] for pricing (be sure to provide your manuscript number and title). Alternatively, you should make your own arrangements to improve the language quality and provide details in your response letter. – PeerJ Staff

Reviewer 1 ·

Basic reporting

The new version of the manuscript address all issues raised in the previous review.

Experimental design

OK

Validity of the findings

OK

Additional comments

N.A.

Cite this review as

---

## Round 0.4 · accepted · Accept

The authors addressed all the technical concerns from the reviewers and fixed the main typesetting issues.